# Impact of human mobility and networking on spread of COVID-19 at the time of the 1st and 2nd epidemic waves in Japan: An effective distance approach

**Yasuhiro Nohara** [1] *, **Toshie Manabe** [2,3]

**1** Utsunomiya University Center for Regional Design, Utsunomiya city, Tochigi, Japan, **2** Nagoya City University Graduate School of Medicine, Nagoya City, Aichi, Japan, **3** Nagoya City University West Medical Center, Nagoya City, Aichi, Japan

* nohara@cc.utsunomiya-u.ac.jp

## Abstract

### Background

The influence of human mobility to the domestic spread of COVID-19 in Japan using the approach of effective distance has not yet been assessed.

### Methods

We calculated the effective distance between prefectures using the data on laboratory-confirmed cases of COVID-19 from January 16 to August 23, 2020, that were times in the 1st and the 2nd epidemic waves in Japan. We also used the aggregated data on passenger volume by transportation mode for the 47 prefectures, as well as those in the private railway, bus, ship, and aviation categories. The starting location (prefecture) was defined as Kanagawa and as Tokyo for the 1st and the 2nd waves, respectively. The accuracy of the spread models was evaluated using the correlation between time of arrival and effective distance, calculated according to the different starting locations.

### Results

The number of cases in the analysis was 16,226 and 50,539 in the 1st and 2nd epidemic waves, respectively. The relationship between arrival time and geographical distance shows that the coefficient of determination was $R^2 = 0.0523$ if geographical distance $D_{geo}$ and time of arrival $T_a$ set to zero at Kanagawa and was $R^2 = 0.0109$ if $D_{geo}$ and $T_a$ set to zero at Tokyo. The relationship between arrival time and effective distance shows that the coefficient of determination was $R^2 = 0.3227$ if effective distance $D_{eff}$ and $T_a$ set to zero at Kanagawa and was $R^2 = 0.415$ if $D_{eff}$ and time of arrival $T_a$ set to zero at Tokyo. In other words, the effective distance taking into account the mobility network shows the spatiotemporal characteristics of the spread of infection better than geographical distance. The correlation of arrival time to effective distance showed the possibility of spreading from multiple areas in

**Data Availability Statement:** All relevant data are within the manuscript and its Supporting Information files. · Supporting Information files. · Geospatial Information Authority of Japan. Ministry of Land, Infrastructure, Transport and Tourism.

[Area by prefecture, city, ward, town, and village]. [cited 2020 Sept 29]. Available from https://www. gsi.go.jp/KOKUJYOHO/MENCHO-title.htm. Japanese. Ministry of Land, Infrastructure, Transport and Tourism of Japan. [Freight/ passenger area flow survey in 2016]. [cited 2020 Sept 29]. Available from https://www.mlit.go.jp/k-toukei/kamoturyokakutiikiryuudoutyousa.html. Japanese. Ministry of Land, Infrastructure, Transport and Tourism of Japan. ['Kokudo Suuchi', the GIS data service of the Ministry of Land, Infrastructure, Transport and Tourism of Japan]. [cited 2020 Sept 29]. Available from https://nlftp. mlit.go.jp/ksj/gml/datalist/KsjTmplt-S05-d-v2_2. html. Japanese.

**Funding:** This study was supported by a grant from the Japan Science and Technology (JST) Mirai Program (#20345310). The funder had no role in the design, methods, participant recruitment, data collection, analysis, or preparation of the paper.

**Competing interests:** The authors have declared that no competing interests exist.

the 1st epidemic wave. On the other hand, the correlation of arrival time to effective distance showed the possibility of spreading from a specific area in the 2nd epidemic wave.

## Conclusions

The spread of COVID-19 in Japan was affected by the mobility network and the 2nd epidemic wave is more affected than those of the 1st epidemic. The effective distance approach has the impact to estimate the domestic spreading COVID-19.

## Introduction

Severe acute respiratory syndrome coronavirus 2 (SARS-CoV-2) was first identified in December 2019 in Wuhan city, Hubei province, China, and has since become a worldwide pandemic [1, 2]. In Japan, the first confirmed case of coronavirus disease (COVID-19) was reported in January 15, 2020, in Kanagawa prefecture [3] and was a traveler returning from Wuhan [3]. Subsequently, laboratory-confirmed cases of COVID-19 were reported sporadically, and then increased gradually from the middle of February. The number of cases rapidly increased from approximately the end of March, and it leaded the first epidemic wave of COVID-19 in Japan. The Japanese government declared a state of emergency on April 7, 2020 [4] and many restrictions to reduce COVID-19 transmission have been taken, including the avoidance of unnecessary travel and limits on traveling across regional borders between prefectures. By the time the state of emergency was cancelled on March 25, 2020, a total of 16,581 COVID-19 cases (and 830 deaths) had been reported [5]. After the cancellation, restrictions were gradually relaxed and movements gradually increased, although people continued to wear masks and take the socially distance. For example, interprefecture travel increased and trains and railway stations became busier, particularly in urban areas. At approximately the end of June, the number of daily cases increased and a second epidemic wave was identified and was characterized by a higher number of cases than the first epidemic wave. The cumulative number of confirmed COVID-19 cases was 67,264 as of August 30, 2020 [6].

Japan has well-developed public transportation systems, such as trains, buses, and airlines, and people can easily move from place to place for business, pleasure, and other activities. COVID-19 transmission routes identified so far include sustained human-to-human transmission [7–9], transmission from an asymptomatic patient [10], and transmission from a pre-symptomatic patient [11]. It is likely that human mobility has had a strong effect on the spread of COVID-19 in Japan.

The concept of effective distance was introduced by Brockman and Helbing in 2013, using data from the influenza H1N1pdm09 pandemic and the outbreak of severe acute respiratory syndrome (SARS) in 2002–2003 SARS outbreak [12]. The concept is based on the idea that places with a dense flow of traffic between them should be effectively closer in a plausible map layout, and places with little traffic between them should be further apart [12]. Studies that have estimated the effective distance indicate that travel restrictions and international airline suspensions have contributed to the spread of COVID-19 [13, 14]. In addition, currently, there are some reports that confronted the effect of different mobility data, including people flow statistics, on the spatiotemporal distributions of SARS-CoV2 at sub-national level [15–17]. Thus, we hypothesized that the number of people traveling between Japanese cities may also affect the domestic spread of COVID-19. The effective distance may need to incorporate an estimate of the spread of COVID-19 to measure further domestic outbreaks.

The aim of the present study was to elucidate how the mobility of people in Japan affects the spread of COVID-19, and the impact of the approach of effective distance to estimate the spread of outbreaks of COVID-19 in Japan. The findings may contribute to further targeted control measures for COVID-19.

## Materials and methods

### Data

Japan is divided into 47 prefectures, with a population of approximately 125 million people [18] and 378 square kilometers of land [19].

Daily data on laboratory-confirmed cases of COVID-19 from January 16 to August 30, 2020, were obtained from publicly available situation reports on prefecture websites (S1 Table).

For mobility network data, we used the aggregated data on passenger volume by transportation mode for the 47 prefectures contained in the freight/passenger area flow survey conducted by the Ministry of Land, Infrastructure, Transport and Tourism of Japan issued in 2016 [20]. The data include the volume of passengers in the private railway, bus, ship, and aviation categories. The dataset counts all transport personnel across other prefectures in one year. It is the OD amount of passenger transport personnel between regions. For example, if the departure point is Tokyo and the arrival point is Osaka, all boarding / alighting personnel by prefecture on the route are counted. These data set was obtained from the "Kokudo Suchi," the geographic information systems (GIS) data service of the Ministry of Land, Infrastructure, Transport and Tourism of Japan [21]. The mobility network diagram of this data consists of 47 nodes (i.e., prefectures) and 1,907 edges connecting prefectures. The value of transport volume excluding the edges is zero. The weight of each edge represents passenger volume between two nodes on all types of transportation (S1 Fig). Although the mobility network data were not generated during the COVID-19 epidemic, the probability of occurrence of movement between prefectures was assumed to be constant.

### Effective distance

To assess the probability of COVID-19 spread within Japan, we calculated the effective distance between prefectures using the mobility network data. Previous studies have shown that effective distance, rather than geographical distance, can predict the arrival time of a virus. This metric was therefore used to identify the starting point of the virus spatial diffusion process.

The basic principle is that despite the structural complexity of the underlying network and the multiplicity of paths, the dynamic process is dominated by a set of most probable paths that can be derived from the connectivity matrix $P$, weighted by passenger volume. The effective distance $d_{ij}$ between the $ith$ prefecture and the $jth$ prefecture, which are directly connected, is defined as $d_{ij} = 1 - ln\,(P_{ij})$. $P_{ij}$ is the transition probability between prefectures. Moreover, the effective distance between an arbitrary reference prefecture and another prefecture in the network is calculated from the minimum of all possible paths.

### Definition of dates and areas for the 1st and 2nd epidemic waves in Japan

During the observational period, the cutoff date between the 1st and the 2nd epidemic peaks was set at May 25, 2020, which was the date the state of emergency was cancelled. This date also presented the lowest number of daily cases between the 1st and 2nd epidemic peaks.

Therefore, in this study, the 1ˢᵗ and 2ⁿᵈ epidemic peaks in Japan were defined as from January 16, 2020, to May 24, 2020, and from May 25 to August 30, 2020, respectively.

The arrival time of infectious disease was set according to the following original criteria. The time of arrival of the 1ˢᵗ epidemic wave was simply set as the day when the prefecture first issued a report of an infected person (S2 Table). The arrival time of the 2ⁿᵈ epidemic wave was simply set as the first day of the period in which the number of infected persons continued to rise for 1 week or more. The duration criterion of 1 week or more was chosen so that outbreaks with some cohesion (i.e., an epidemic wave) were selected rather than sporadic outbreaks. In the 2ⁿᵈ epidemic wave, it was confirmed that the number of cases continued to rise for more than 1 week in 39 prefectures. The other eight prefectures were not included in the analysis, as they did not show a 2ⁿᵈ epidemic wave.

The starting location (prefecture) of the 1ˢᵗ epidemic wave was defined as Kanagawa prefecture, where the first laboratory-confirmed case of COVID-19 in Japan was identified [3]. The starting location of the 2ⁿᵈ epidemic wave was defined as Tokyo, which had one of the highest prevalence of COVID-19 cases at the cutoff week between the 1ˢᵗ and the 2ⁿᵈ epidemic waves (and also in the following weeks).

## Evaluation of the accuracy of the spread models

The accuracy of the spread models was evaluated using the correlation between time of arrival and effective distance, calculated according to the different starting locations all prefectures. Hokkaido, Chiba, Tokyo, Kanagawa, Aichi, Osaka, Fukuoka, and Okinawa that have the major international airports serving international flights to more than fifteen destination cities. The other seven locations are prefectures in which the major international airports are located, but the number of destination cities were less than eight. The first laboratory-confirmed case of COVID-19 in Japan was identified in Kanagawa [3].

## Geographical distance

We calculated the geographical distance between prefectures to confirm the effectiveness of the effective distance measure. Geographical distance was measured using a geographic information system (ArcMap 10.7.1, ESRI Japan, Tokyo) that calculated the linear distance between the prefectural capitals.

## Results

### Number of cases and arrival time for each epidemic wave

The number of cases in the analysis data was 16,226 in the 1ˢᵗ epidemic wave and 50,539 in the 2ⁿᵈ epidemic wave. The 1ˢᵗ epidemic wave of infection was conspicuous in large cities in Hokkaido, Tokyo, Aichi, Osaka, and neighboring prefectures (Fig 1A). Furthermore, infection spread was observed in the local cities of Ishikawa and Toyama. In the 2ⁿᵈ epidemic wave (Fig 1B), as in the 1ˢᵗ epidemic wave, the number of cases increased in large cities and neighboring prefectures. Additionally, the infection spread to some different locations (e.g., Miyazaki, Kagoshima, and Okinawa) compared with the 1ˢᵗ epidemic wave.

Time of arrival of each epidemic wave is shown in (Fig 2A and 2B). In the 1ˢᵗ epidemic wave, the infection began to spread sporadically in geographically distant areas such as around Tokyo, around Osaka, and Hokkaido (Fig 2A). In contrast, in the 2ⁿᵈ epidemic wave, the starting point was limited to Tokyo, Kanagawa, and Hokkaido (Fig 2B). In either case, it is unlikely that the infection spread based on geographical proximity.

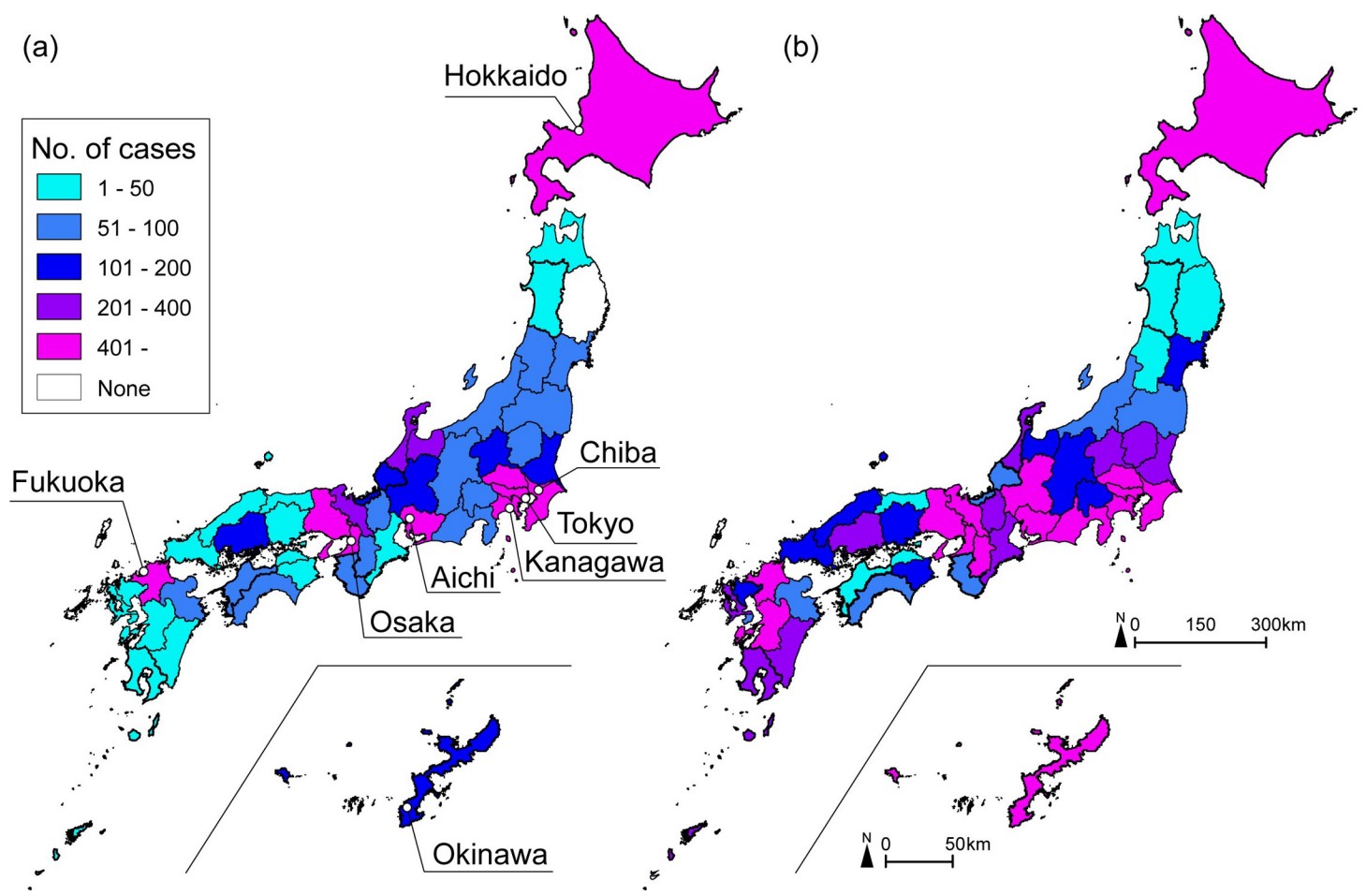

**Fig 1. No. of cases in 1$^{st}$ epidemic wave and 2$^{nd}$ epidemic wave.** (a) No. of cases at 1$^{st}$ epidemic wave (January 16 to May 24). (b) No. of cases at 2$^{nd}$ epidemic wave (May 25 to August 30). Total no. of cases in each prefecture is shown in S1 Table.

### Effective distance shows the spatiotemporal characteristics of infection spread

The relationship between arrival time and geographical distance is shown in (Fig 3A and 3B). This is based on the prefecture where the first case was confirmed in the 1st and 2nd epidemic waves. Fig 3A shows geographical distance $D_{geo}$ and time of arrival $T_a$ set to zero at Kanagawa; the coefficient of determination was R$^2$ = 0.0523 (p-value > 0.05). Fig 3B shows geographical distance $D_{geo}$ and time of arrival $T_a$ set to zero at Tokyo; the coefficient of determination was R$^2$ = 0.0109 (p-value > 0.05). The relationship between arrival time and effective distance is shown in (Fig 3C and 3D). Fig 3C shows effective distance $D_{eff}$ and time of arrival $T_a$ set to zero at Kanagawa, where reports of infected people were first issued in Japan. The coefficient of determination was R$^2$ = 0.3227 (p-value < 0.05). Fig 3D shows effective distance $D_{eff}$ and time of arrival $T_a$ set to zero at Tokyo, which had the highest number of infected people as of May 25. The coefficient of determination was R$^2$ = 0.415 (p-value < 0.05). The model for the 2$^{nd}$ epidemic wave fits the linear model better than the model for the 1$^{st}$ epidemic wave. Compared with the model for arrival time against effective distance in Fig 2, the arrival time and geographical distance were independent of each other. In other words, the effective distance

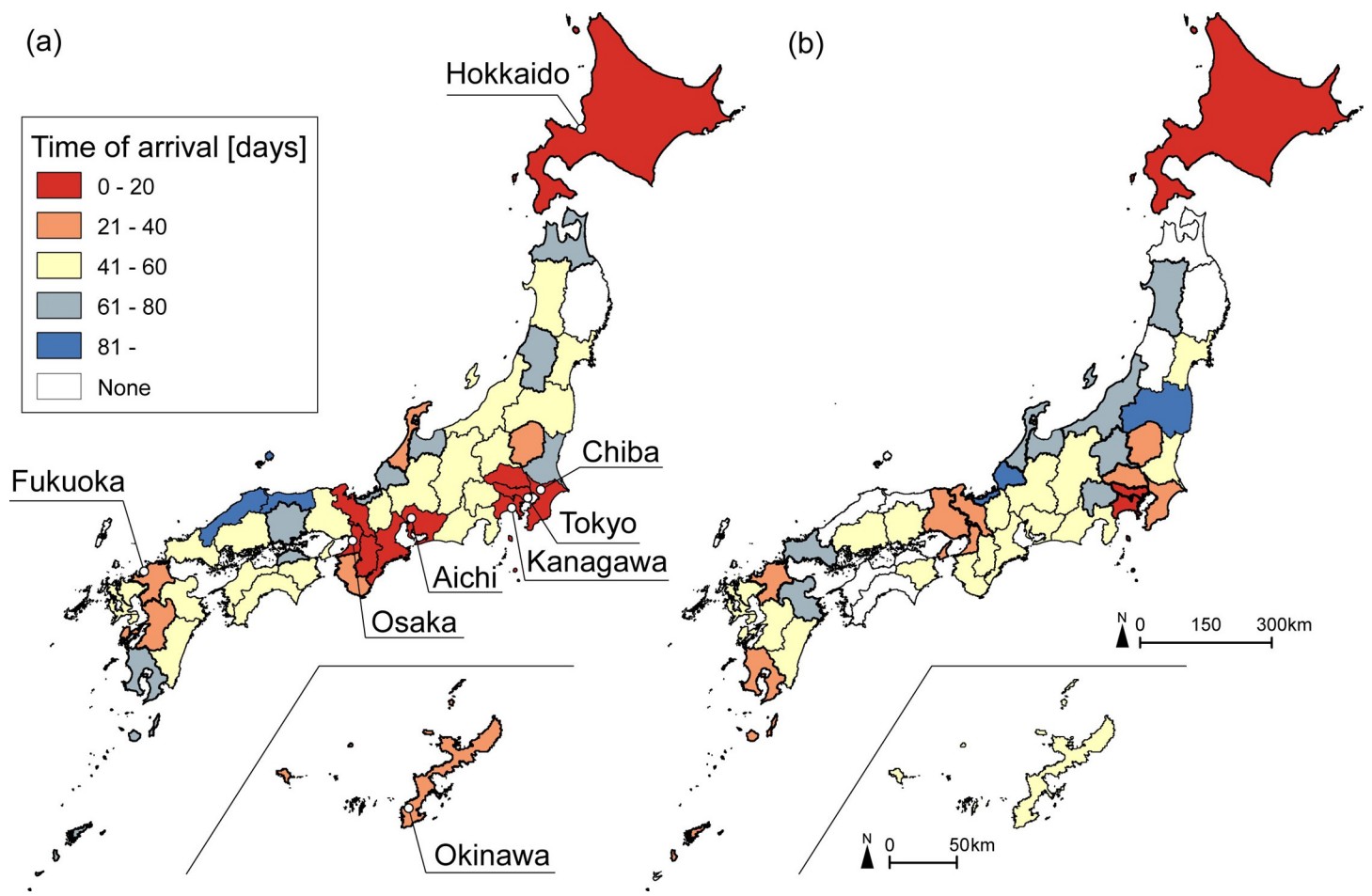

**Fig 2. Time of arrival of 1ˢᵗ epidemic wave and 2ⁿᵈ epidemic wave.** (a) No. of cases at 1ˢᵗ epidemic wave (January 16 to May 24). (b) No. of cases at 2ⁿᵈ epidemic wave (May 25 to August 30). Time of arrival of the virus in each prefecture is shown in S2 Table.

taking into account the mobility network is a better representation of the spatiotemporal characteristics of the spread of infection.

## Correlation between time of arrival and effective distance

Table 1 shows the Spearman correlation coefficients for multiple starting points of effective distances. In the 1ˢᵗ epidemic wave, the correlation was strongest when starting from Mie (r = 0.661), not Kanagawa, where the first case was confirmed in Japan. The prefecture with the second strongest correlation is Aichi (r = 0.571), and the third is Kanagawa (r = 0.568). Mie is adjacent to Aichi, where the major international airports are located, and these areas may have become seeds for transmission routes. On the other hand, Kanagawa, where the first case was confirmed in Japan, may also be the seed of the infection route. The two regions are geographically separated and it is difficult to estimate the starting point of the spread of the infection. It is also possible that the infection propagated from two geographically independent regions as a starting point. In the 2ⁿᵈ epidemic wave, the correlation was strongest when starting from Tokyo (r = 0.644). The prefecture with the second strongest correlation is Kanagawa (r = 0.629), and the third is Chiba (r = 0.602), which are adjacent to each other across Tokyo. In other words, it is possible that these areas were used as seeds to spread the infection. What

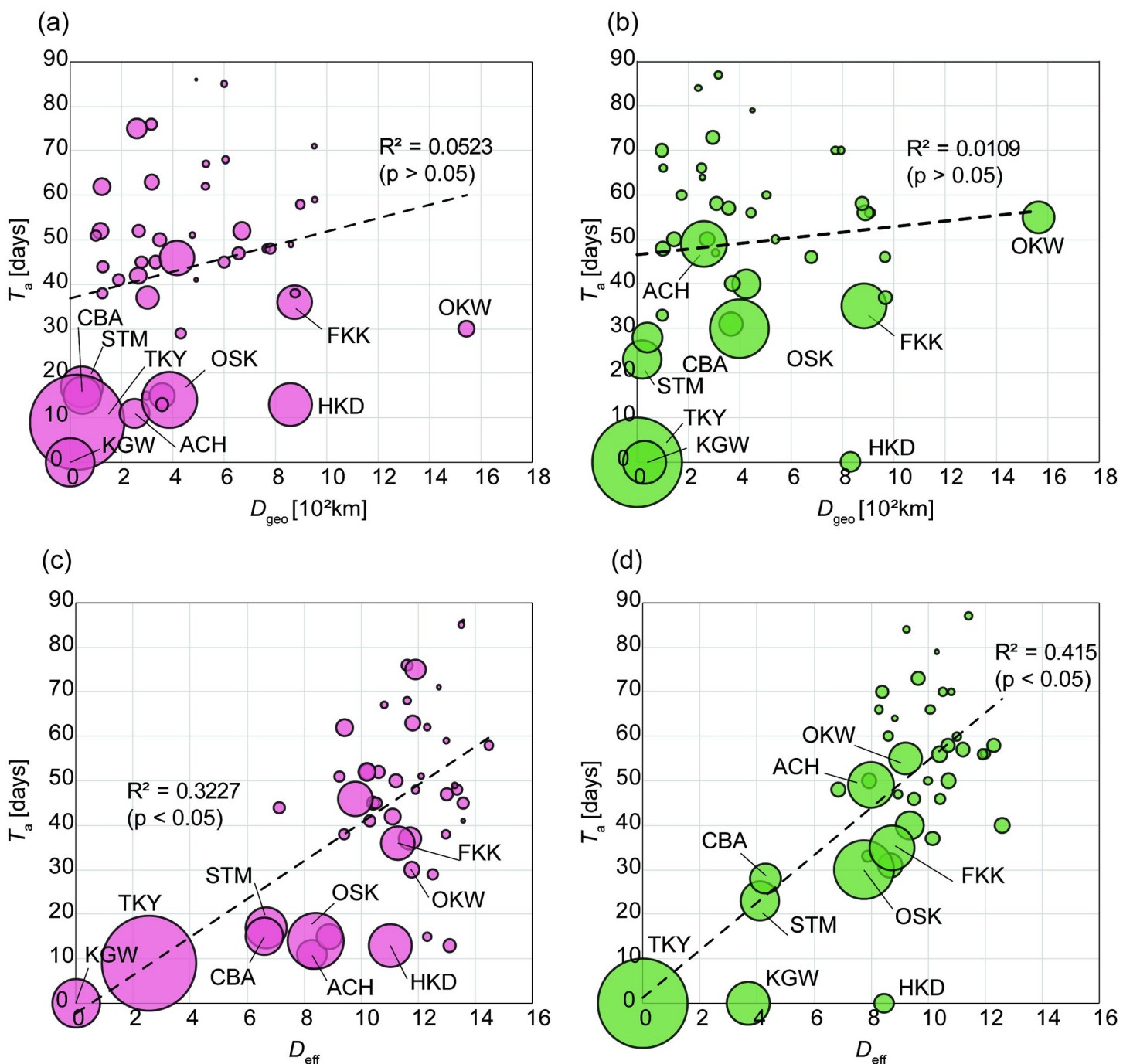

**Fig 3. Linear relationship between time of arrival $T_a$ and effective distance $D_{eff}$.** (a and b) Linear relationship between time of arrival $T_a$ and geographical distance $D_{geo}$ at 1st epidemic wave from Kanagawa (a) and at 2nd epidemic wave from Tokyo (b). (c and d) Linear relationship between time of arrival $T_a$ at 1st epidemic wave from Kanagawa (c) and at 2nd epidemic wave from Tokyo (d). HKD: Hokkaido, CBA: Chiba, TKY: Tokyo, KGW: Kanagawa, ACH: Aichi, OSK: Osaka, FKK: Fukuoka, OKW: Okinawa, STM: Saitama. The size of the bubble represents the sum of the number of cases in each prefecture.

is important here is that while it is difficult to estimate the origin of the spread of infection in the 1st epidemic wave, the 2nd epidemic wave can be estimated to have propagated based on the effective distance considering the mobility network with a specific area as the origin.

**Table 1. Correlation between time of arrival and effective distance.**

| Prefecture name | The Spearman's correlation coefficients at 1st epidemic wave | The Spearman's correlation coefficients at 2nd epidemic wave |
|---|---|---|
| Hokkaido* | 0.482 | 0.578 |
| Aomori | 0.334 | 0.419 |
| Iwate | 0.348 | 0.369 |
| Miyagi | 0.315 | 0.328 |
| Akita | 0.334 | 0.271 |
| Yamagata | 0.275 | 0.308 |
| Fukushima | 0.290 | 0.245 |
| Ibaraki | 0.387 | 0.449 |
| Tochigi | 0.356 | 0.378 |
| Gunma | 0.370 | 0.251 |
| Saitama | 0.386 | 0.442 |
| Chiba* | 0.495 | 0.602 |
| Tokyo* | 0.488 | 0.644 |
| Kanagawa* | 0.568 | 0.629 |
| Niigata | 0.352 | 0.252 |
| Toyama | 0.263 | 0.191 |
| Ishikawa | 0.339 | 0.150 |
| Fukui | 0.279 | 0.097 |
| Yamanashi | 0.438 | 0.297 |
| Nagano | 0.345 | 0.215 |
| Gifu | 0.401 | 0.170 |
| Shizuoka | 0.502 | 0.437 |
| Aichi* | 0.571 | 0.347 |
| Mie | 0.661 | 0.303 |
| Shiga | 0.358 | 0.235 |
| Kyoto | 0.477 | 0.321 |
| Osaka* | 0.515 | 0.401 |
| Hyogo | 0.354 | 0.460 |
| Nara | 0.566 | 0.341 |
| Wakayama | 0.550 | 0.312 |
| Tottori | -0.007 | 0.365 |
| Shimane | 0.010 | 0.410 |
| Okayama | 0.094 | 0.397 |
| Hiroshima | 0.163 | 0.386 |
| Yamaguchi | 0.064 | 0.229 |
| Tokushima | 0.269 | 0.468 |
| Kagawa | 0.177 | 0.307 |
| Ehime | 0.260 | 0.347 |
| Kochi | 0.258 | 0.422 |
| Fukuoka* | 0.243 | 0.370 |
| Saga | 0.211 | 0.347 |
| Nagasaki | 0.200 | 0.341 |
| Kumamoto | 0.250 | 0.312 |
| Oita | 0.239 | 0.251 |
| Miyazaki | 0.329 | 0.416 |
| Kagoshima | 0.216 | 0.393 |

(*Continued*)

**Table 1.** (Continued)

| Prefecture name | The Spearman's correlation coefficients at 1st epidemic wave | The Spearman's correlation coefficients at 2nd epidemic wave |
|---|---:|---:|
| Okinawa* | 0.539 | 0.567 |

*; Hokkaido, Chiba, Tokyo, Kanagawa, Aichi, Osaka, Fukuoka, and Okinawa that have the major international airports serving international flights to more than fifteen destination cities.

## Discussion

The effective distance taking into account the mobility network shows the spatiotemporal characteristics of the spread of infection better than geographical distance. The correlation of arrival time to effective distance showed the possibility of spreading from multiple areas in the 1st epidemic wave. On the other hand, the correlation of arrival time to effective distance showed the possibility of spreading from a specific area in the 2nd epidemic wave. In the 1st epidemic wave, the possibility of multiple linear models starting from multiple regions and it was difficult to estimate the starting point of the infection spread. The infection may have spread sporadically owing to the effect of overseas travelers. In contrast, in the 2nd epidemic wave, the effect of overseas travelers was not taken into consideration under border control, and the influence of domestic movement may have been noticeable.

In Japan, control measures are generally adopted on a prefecture-by-prefecture basis during an infectious disease epidemic. Therefore, the timing and period of control measures such as movement restrictions and lockdown differ from prefecture to prefecture. However, in prefectures such as Tokyo, Saitama, Chiba, and Kanagawa, which are geographically close to each other and have a short effective distance, integrated measures are necessary

Additionally, as noted in previous research [14], the assessment of regional vulnerabilities by combining effective distance and analysis of medical resources, knowing in advance which regions may be most affected, might have allowed authorities to opt for preemptive differential investments in these regions. However, it should be noted that in some areas, such as Hokkaido, the infection spread independently. Such unique areas should consider control measures in smaller units.

Our study had some limitations. We used 2016 data on mobility networks and passenger volume, and the transition probability between prefectures was assumed to be constant. It is unclear how domestic movements have changed as a result of COVID-19, and it is necessary to improve the accuracy of the transition. We therefore propose further research to calculate the effective distance based on real-time people-flow statistical data from such sources as mobile devices and Wi-Fi. The results for the relationship between effective distance and arrival time changed according to the setting of arrival time for each epidemic wave. There are no clear criteria for the arrival time of infectious disease in each region. In particular, it is difficult to determine the arrival time of the 2nd epidemic wave. There are various possible measures of arrival time, such as the date of the report of the first infected person, a report of continual infections over a specific time period, and the point at which the number of infected persons reaches a certain number. Although there is a model that estimates the period from occurrence of the 1st epidemic wave to occurrence of the 2nd epidemic wave [22], no previous studies seem to discuss the starting point of the 2nd epidemic wave. Similarly, to the best of our knowledge, previous studies using the theory of effective distance focus on the 1st epidemic [12–14]. In this study, one simple method was used to set the criteria. Future studies should address various potential criteria.

## Supporting information

**S1 Fig. Complete mobility network diagram in Japan in 2016.**
(TIF)

**S1 Table. Total number of cases in each prefecture.**
(DOCX)

**S2 Table. Infection arrival time for each prefecture.**
(DOCX)

## Acknowledgments

The authors would like to express our sincere appreciation to Sarika Nakamura for her general assistance.

## Author Contributions

**Conceptualization:** Yasuhiro Nohara, Toshie Manabe.

**Data curation:** Yasuhiro Nohara, Toshie Manabe.

**Formal analysis:** Yasuhiro Nohara.

**Funding acquisition:** Toshie Manabe.

**Investigation:** Yasuhiro Nohara, Toshie Manabe.

**Methodology:** Yasuhiro Nohara.

**Project administration:** Yasuhiro Nohara.

**Resources:** Yasuhiro Nohara, Toshie Manabe.

**Software:** Yasuhiro Nohara.

**Supervision:** Yasuhiro Nohara.

**Validation:** Yasuhiro Nohara.

**Visualization:** Yasuhiro Nohara.

**Writing – original draft:** Yasuhiro Nohara.

**Writing – review & editing:** Yasuhiro Nohara, Toshie Manabe.

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
