## [Decision Letter · Decision Letter 0]

8 Apr 2022

PONE-D-21-37364Impact of human mobility and networking on spread of COVID-19 at the time of the 1st and 2nd epidemic waves in Japan: an effective distance approachPLOS ONE

Dear Dr. Nohara,

Thank you for submitting your manuscript to PLOS ONE. After careful consideration, we feel that it has merit but does not fully meet PLOS ONE’s publication criteria as it currently stands. Therefore, we invite you to submit a revised version of the manuscript that addresses the points raised during the review process.

We look forward to receiving your revised manuscript.

Kind regards,

Chiara Poletto

Academic Editor

PLOS ONE

Journal Requirements:

Reviewers' comments:

Reviewer's Responses to Questions

**Comments to the Author**

1. Is the manuscript technically sound, and do the data support the conclusions?

Reviewer #1: Yes

Reviewer #2: Yes

2. Has the statistical analysis been performed appropriately and rigorously? 

Reviewer #1: No

Reviewer #2: Yes

3. Have the authors made all data underlying the findings in their manuscript fully available?

Reviewer #1: Yes

Reviewer #2: Yes

4. Is the manuscript presented in an intelligible fashion and written in standard English?

Reviewer #1: Yes

Reviewer #2: Yes

5. Review Comments to the Author

Reviewer #1: The authors provide an application of the popular effective distance metric from Brockmann and Helbing (2013) to study the arrival times of SARS-CoV2 in Japan at prefectures level during the 1st and 2nd wave of infections. Their results suggest that Tokyo and Kanagawa prefectures were the starting geographical regions for the spreading of Covid-19 through the country.

- a detailed description of mobility data is missing, to what year to the mobility data refer to? What do they represent? Is their representativity similar? the authors mix together different types of mobility data, but it is not clear if their representativity is the same, if they cover the same number of days. To mix different sources of mobility data is a sensible process, which can determine an overrepresentation of some modes of transportation over the others.

- how are the trips considered in the dataset? If a person travels from Tokyo to Osaka while stopping through all the prefectures in the middle, are all the steps considered or you only get the number of people travelling directly from Tokyo to Osaka? For example, is any stop time considered in order to separate trips? Or the dataset only accounts for the origin and destination showed on the purchased tickets?

- The authors say “The accuracy of the spread models was evaluated using the correlation between time of arrival and effective distance, calculated according to the different starting locations: Hokkaido, Chiba, Tokyo, Kanagawa, Aichi, Osaka, Fukuoka, and Okinawa that have the major international airports serving international flights to more than fifteen destination cities. The other seven locations are prefectures in which the major international airports are located, but the number of destination cities were less than eight. “

The authors said that they had 47 prefectures in the dataset, but the models are tested taking as seeding locations only 8 prefectures, whereas later they say “the other 7 locations are prefectures in which the major international airports are located”. I am confused at this point on how many administrative areas are they considering in their model. The model is tested only on a subset of the total dataset and the destinations seem reduced to 15 out of 47. I do not understand the logic of this choice. Why not considering all the prefectures are starting locations and destination locations? The ones that you listed are surely those with the biggest airports, but you said that the mobility data includes all modes of transportations, such as trains, bus, etcetera. Hence finally all prefectures could be tested as starting (seeding) points, possibly showing similar correlation scores if they are strongly connected.

- the authors may want to discuss the role of cases underdetection and the different testing capacities implemented in the different prefectures, which may have affected the detection of the first infected person in the area, and hence the time of arrival of the virus in the geographical area. Figure 2 shows arrival times that differ by few days between all the prefectures taken into account, small differences in cases underdetection would strongly affect the correlation, and indeed points in Fig.2a look very dispersed. The p-value for these correlation should be reported in the picture.

-Why not considering the date of reaching a certain amount of daily case incidence instead of using such a sensible measure like the arrival time of the first case (see for example Ref.[1] in assessing the spreaders role)? The role of asymptomatic infections has been already widely studied in the case of SARS-CoV2, single cases do not necessarily trigger an outbreak, e.g. many reported individual and isolated cases in Europe were reported in January 2020 without starting any local epidemic. Moreover, long times of disease incubation hinders the detection of the first infected patients in their actual areas of arrival from international travel. So what is the logic of relying on such a definition from a public health point of view for SARS-CoV2, when the detection of a single case are not necessarily representative for the triggering of an outbreak in the area?

- As a more general reasoning, I would like to know what is the authors’ thought on the following problem. The model is tested only on few starting prefectures, which surely are those who are the most connected ones in terms of international flights, however they also correspond to the most populated areas in which testing capacity may be higher than other peripheral prefectures, and hence outbreak detection is more efficient. What would be the probability of correctly detecting the arrival time of an infected patient in a peripheral area that has a poor testing capacity? In brief, would we be able through the effective distance model to correctly assess the start of an outbreak from a peripheral area, given these circumstances? Or would we always detect the earliest arrival time in the most tested prefectures and mobility hubs as the result of a demographic and testing capacity bias?

- In this sense, how can be we sure that Tokyo and Kanagawa are effectively the starting prefectures of the 1st and 2nd waves, only from the correlation of effective distance and arrival times, given the possible confounder effects represented by population density? Infrastructures are planned on the basis of gravity (or radiation) models that take into account origin and destination populations, so effective distance from mobility hubs at prefectures level would reflect the population hierarchy. Given these confounding factors, how can we be sure that this model provides evidence to say that Tokyo and Kanagawa were the seeding prefectures in Japan?

- the introduction on previous works and state of the art on mobility and epidemics is insufficient. There are many many works that lately confronted the effect of different mobility data on the spatiotemporal invasion of SARS-CoV2 at sub-national level that need to be correctly referenced.

See for example these three papers and their references to build a more general overview in the introduction:

[1] Mazzoli, Mattia, et al. "Interplay between mobility, multi-seeding and lockdowns shapes COVID-19 local impact." PLoS computational biology 17.10 (2021): e1009326.

[2] Kraemer, Moritz UG, et al. "Spatiotemporal invasion dynamics of SARS-CoV-2 lineage B. 1.1. 7 emergence." Science 373.6557 (2021): 889-895.

[3] Kraemer, Moritz UG, et al. "The effect of human mobility and control measures on the COVID-19 epidemic in China." Science 368.6490 (2020): 493-497.

Reviewer #2: This work analyses how the human mobility in Japan may have affected the spreading of SARS-CoV-2 in the first two waves. It does so by computing the correlation between the effective distance of a prefecture from the starting point of the epidemic in Japan and the time of first-arrival of the epidemic in that prefecture.

This idea to use the concept of effective distance to gain insights on the epidemics is not new, but to the best of my knowledge this is the first time that it has been used for the public transportation network of Japan, making the paper original worth publishing.

I only have one doubt: throughout the paper it is mentioned various times that an higher Spearman coefficient indicates a better fit with the linear model. However the Spearman coefficient measures how closely the data follow a monotonic function, which in general may not be linear. So in my opinion the paper would gain in rigorousness if this problem was addressed or if instead of “linear model” the authors used a more generic term like “positive correlation”.

6. PLOS authors have the option to publish the peer review history of their article (what does this mean?). If published, this will include your full peer review and any attached files.

Reviewer #1: No

Reviewer #2: No

---

## [Author Response · Author response to Decision Letter 0]

24 Jun 2022

Point-by-point response to reviewers’ comments

We would like to express our sincere appreciation to the editor and reviewers for their kind comments and suggestions regarding our manuscript. We have included our responses to the reviewers’ comments herein, for ease of reference. For clarity, the comments are shown in blue and bold font and our responses are shown in black and regular font.

Reviewer reports:

Reviewer 1: 

- The authors provide an application of the popular effective distance metric from Brockmann and Helbing (2013) to study the arrival times of SARS-CoV2 in Japan at prefectures level during the 1st and 2nd wave of infections. Their results suggest that Tokyo and Kanagawa prefectures were the starting geographical regions for the spreading of Covid-19 through the country.

- a detailed description of mobility data is missing, to what year to the mobility data refer to? What do they represent? Is their representativity similar? the authors mix together different types of mobility data, but it is not clear if their representativity is the same, if they cover the same number of days. To mix different sources of mobility data is a sensible process, which can determine an overrepresentation of some modes of transportation over the others.

Thank you very much for taking your time for reviewing our manuscript and providing valuable advice and suggestions. 

Regarding the mobility data, the issued year and some additional descriptions have been included in the method section. We have rewrite them for more clearly understanding.

Also, we mentioned about using the different sources of mobility data in 2016 as the study limitation in the discussion section.

P. 6-7; Lines 52-60.

For mobility network data, we used the aggregated data on passenger volume by transportation mode for the 47 prefectures contained in the freight/passenger area flow survey conducted by the Ministry of Land, Infrastructure, Transport and Tourism of Japan issued in 2016 [20]. The data include the volume of passengers in the private railway, bus, ship, and aviation categories. The dataset counts all transport personnel across other prefectures in one year. It is the OD amount of passenger transport personnel between regions. For example, if the departure point is Tokyo and the arrival point is Osaka, all boarding / alighting personnel by prefecture on the route are counted. These data set was obtained from the “Kokudo Suchi,” the geographic information systems (GIS) data service of the Ministry of Land, Infrastructure, Transport and Tourism of Japan [21].

P. 17-18; Lines 228-230.

We used 2016 data on mobility networks and passenger volume, and the transition probability between prefectures was assumed to be constant.

- how are the trips considered in the dataset? If a person travels from Tokyo to Osaka while stopping through all the prefectures in the middle, are all the steps considered or you only get the number of people travelling directly from Tokyo to Osaka? For example, is any stop time considered in order to separate trips? Or the dataset only accounts for the origin and destination showed on the purchased tickets?

The dataset counts all transport personnel across other prefectures in one year. It is the OD amount of passenger transport personnel between regions. For example, if the departure point is Tokyo and the arrival point is Osaka, all boarding / alighting personnel by prefecture on the route are counted. We added these detailed explanation in the materials and methods section in the text.

P. 6-7; Lines 55-59.

The data include the volume of passengers in the private railway, bus, ship, and aviation categories. The dataset counts all transport personnel across other prefectures in one year. It is the OD amount of passenger transport personnel between regions. For example, if the departure point is Tokyo and the arrival point is Osaka, all boarding / alighting personnel by prefecture on the route are counted.

- The authors say “The accuracy of the spread models was evaluated using the correlation between time of arrival and effective distance, calculated according to the different starting locations: Hokkaido, Chiba, Tokyo, Kanagawa, Aichi, Osaka, Fukuoka, and Okinawa that have the major international airports serving international flights to more than fifteen destination cities. The other seven locations are prefectures in which the major international airports are located, but the number of destination cities were less than eight. “

The authors said that they had 47 prefectures in the dataset, but the models are tested taking as seeding locations only 8 prefectures, whereas later they say “the other 7 locations are prefectures in which the major international airports are located”. I am confused at this point on how many administrative areas are they considering in their model. The model is tested only on a subset of the total dataset and the destinations seem reduced to 15 out of 47. I do not understand the logic of this choice. Why not considering all the prefectures are starting locations and destination locations? The ones that you listed are surely those with the biggest airports, but you said that the mobility data includes all modes of transportations, such as trains, bus, etcetera. Hence finally all prefectures could be tested as starting (seeding) points, possibly showing similar correlation scores if they are strongly connected.

In this article, considering the transmission route from abroad, we focused only on the eight prefectures where the major international airports are located. As you point out, we need to show the correlation between arrival dates and effective distances in 47 prefectures. Therefore, Table 1 was revised to show the correlation between 47 prefectures (P. 14-16; Lines 198-201). In response to this revision, the text has been revised as follows.

P. 10; Lines 109-111.

The accuracy of the spread models was evaluated using the correlation between time of arrival and effective distance, calculated according to the different starting locations all prefectures.

P. 14; Lines 181-196.

In the 1st epidemic wave, the correlation was strongest when starting from Mie (r =0.661), not Kanagawa, where the first case was confirmed in Japan. The prefecture with the second strongest correlation is Aichi (r = 0.571), and the third is Kanagawa (r = 0.568). Mie is adjacent to Aichi, where the major international airports are located, and these areas may have become seeds for transmission routes. On the other hand, Kanagawa, where the first case was confirmed in Japan, may also be the seed of the infection route. The two regions are geographically separated and it is difficult to estimate the starting point of the spread of the infection. It is also possible that the infection propagated from two geographically independent regions as a starting point. In the 2nd epidemic wave, the correlation was strongest when starting from Tokyo (r = 0.644). The prefecture with the second strongest correlation is Kanagawa (r = 0.629), and the third is Chiba (r = 0.602), which are adjacent to each other across Tokyo. In other words, it is possible that these areas were used as seeds to spread the infection. What is important here is that while it is difficult to estimate the origin of the spread of infection in the 1st epidemic wave, the 2nd epidemic wave can be estimated to have propagated based on the effective distance considering the mobility network with a specific area as the origin.

P. 16; Lines 207-212.

The correlation of arrival time to effective distance showed the possibility of spreading from multiple areas in the 1st epidemic wave and the possibility of spreading from a specific area in the 2nd epidemic wave. In the 1st epidemic wave, the possibility of multiple linear models starting from multiple regions and it was difficult to estimate the starting point of the infection spread.

- the authors may want to discuss the role of cases underdetection and the different testing capacities implemented in the different prefectures, which may have affected the detection of the first infected person in the area, and hence the time of arrival of the virus in the geographical area. Figure 2 shows arrival times that differ by few days between all the prefectures taken into account, small differences in cases underdetection would strongly affect the correlation, and indeed points in Fig.2a look very dispersed. The p-value for these correlation should be reported in the picture.

The significance level based on the calculation of the p-value was added to Fig.3 and text.

P. 12-13; Lines 153-163

Fig 3(a) shows geographical distance Dgeo and time of arrival Ta set to zero at Kanagawa; the coefficient of determination was R2 = 0.0523 (p-value > 0.05). Fig 3(b) shows geographical distance Dgeo and time of arrival Ta set to zero at Tokyo; the coefficient of determination was R2 = 0.0109 (p-value > 0.05). The relationship between arrival time and effective distance is shown in Fig 3(c and d). Fig 3(c) shows effective distance Deff and time of arrival Ta set to zero at Kanagawa, where reports of infected people were first issued in Japan. The coefficient of determination was R2 = 0.3227 (p-value < 0.05). Fig 3(d) shows effective distance Deff and time of arrival Ta set to zero at Tokyo, which had the highest number of infected people as of May 25. The coefficient of determination was R2 = 0.415 (p-value < 0.05).

-Why not considering the date of reaching a certain amount of daily case incidence instead of using such a sensible measure like the arrival time of the first case (see for example Ref.[1] in assessing the spreaders role)? The role of asymptomatic infections has been already widely studied in the case of SARS-CoV2, single cases do not necessarily trigger an outbreak, e.g. many reported individual and isolated cases in Europe were reported in January 2020 without starting any local epidemic. Moreover, long times of disease incubation hinders the detection of the first infected patients in their actual areas of arrival from international travel. So what is the logic of relying on such a definition from a public health point of view for SARS-CoV2, when the detection of a single case are not necessarily representative for the triggering of an outbreak in the area?

As you point out, there is a time lag in inducing infection, and it is not possible to determine the signs of virus arrival from a single case that reaches the area. Since the purpose of this study is to discuss how the infection was reached in 47 prefectures, the time when the infection was first announced in that prefecture is used. In fact, there is no clear standard for arrival time (listed at the end of the sentence).

When performing a detailed infection spread simulation considering the incubation period in a specific area, it is necessary to calculate by focusing on the case incidence rate you propose. The three documents you presented [1-3] performed simulations based on real-time datasets, and provided excellent suggestions for considering the arrival time and the impact of mobility networks. These references were referenced in the Limitation section. After that, I would like to challenge the real data set and complicated condition setting to improve the accuracy.

P. 18; Lines 231-233.

We therefore propose further research to calculate the effective distance based on real-time people-flow statistical data from such sources as mobile devices and Wi-Fi.

- As a more general reasoning, I would like to know what is the authors’ thought on the following problem. The model is tested only on few starting prefectures, which surely are those who are the most connected ones in terms of international flights, however they also correspond to the most populated areas in which testing capacity may be higher than other peripheral prefectures, and hence outbreak detection is more efficient. What would be the probability of correctly detecting the arrival time of an infected patient in a peripheral area that has a poor testing capacity? In brief, would we be able through the effective distance model to correctly assess the start of an outbreak from a peripheral area, given these circumstances? Or would we always detect the earliest arrival time in the most tested prefectures and mobility hubs as the result of a demographic and testing capacity bias?

In response to your suggestions, we have revised Table 1 to show the correlation for 47 prefectures. As shown in Table 1, in the 1st epidemic wave, the correlation coefficient was the highest when Mie prefecture, which has a relatively small population, was used as a seed. In other words, it indicates that the infection may have spread from areas where international flights are not connected. However, unfortunately, the probability of correctly detecting the arrival time is unclear because no clear criteria have been set for the arrival time of the infection. This is the limitation of analysis in this study.

- In this sense, how can be we sure that Tokyo and Kanagawa are effectively the starting prefectures of the 1st and 2nd waves, only from the correlation of effective distance and arrival times, given the possible confounder effects represented by population density? Infrastructures are planned on the basis of gravity (or radiation) models that take into account origin and destination populations, so effective distance from mobility hubs at prefectures level would reflect the population hierarchy. Given these confounding factors, how can we be sure that this model provides evidence to say that Tokyo and Kanagawa were the seeding prefectures in Japan?

In response to your suggestions, we have revised Table 1 to show the correlation for 47 prefectures. The text has been revised accordingly. The possibilities of the prefectures where the 1st epidemic wave started have expanded. However, unfortunately, the probability of correctly detecting the arrival time is unclear because no clear criteria have been set for the arrival time of the infection. This is the limitation of analysis in this study. Therefore, although it is impossible to determine the seeding prefecture, we were able to grasp the tendency of the spread of infection with seeds in a specific area as shown in the text.

- the introduction on previous works and state of the art on mobility and epidemics is insufficient. There are many many works that lately confronted the effect of different mobility data on the spatiotemporal invasion of SARS-CoV2 at sub-national level that need to be correctly referenced.

See for example these three papers and their references to build a more general overview in the introduction:

[1] Mazzoli, Mattia, et al. "Interplay between mobility, multi-seeding and lockdowns shapes COVID-19 local impact." PLoS computational biology 17.10 (2021): e1009326.

[2] Kraemer, Moritz UG, et al. "Spatiotemporal invasion dynamics of SARS-CoV-2 lineage B. 1.1. 7 emergence." Science 373.6557 (2021): 889-895.

[3] Kraemer, Moritz UG, et al. "The effect of human mobility and control measures on the COVID-19 epidemic in China." Science 368.6490 (2020): 493-497.

Thank you very much for your kind suggestion. Per your advice, we referred the issue about three major reports regarding the effect of different mobility data on the spatiotemporal invasion of SARS-CoV2 in the introduction section. Also, we added the comment about the related issue of theses references in the limitation section.

P. 5; Lines 34 - 36. 

In addition, currently, there are some reports that confronted the effect of different mobility data, including people flow statistics, on the spatiotemporal distributions of SARS-CoV2 at sub-national level [15 - 17].

P. 18; Lines 231-233.

We therefore propose further research to calculate the effective distance based on real-time people-flow statistical data from such sources as mobile devices and Wi-Fi. 

 

Reviewer #2:

This work analyses how the human mobility in Japan may have affected the spreading of SARS-CoV-2 in the first two waves. It does so by computing the correlation between the effective distance of a prefecture from the starting point of the epidemic in Japan and the time of first-arrival of the epidemic in that prefecture.

This idea to use the concept of effective distance to gain insights on the epidemics is not new, but to the best of my knowledge this is the first time that it has been used for the public transportation network of Japan, making the paper original worth publishing.

I only have one doubt: throughout the paper it is mentioned various times that an higher Spearman coefficient indicates a better fit with the linear model. However the Spearman coefficient measures how closely the data follow a monotonic function, which in general may not be linear. So in my opinion the paper would gain in rigorousness if this problem was addressed or if instead of “linear model” the authors used a more generic term like “positive correlation”.

As you point out, the analysis should show Spearman's correlation coefficient, not the judgment of compatibility with the linear model. I refrained from using the term linear model in the text as much as possible.

---

## [Decision Letter · Decision Letter 1]

1 Aug 2022

Impact of human mobility and networking on spread of COVID-19 at the time of the 1st and 2nd epidemic waves in Japan: an effective distance approach

PONE-D-21-37364R1

Dear Dr. Nohara,

We’re pleased to inform you that your manuscript has been judged scientifically suitable for publication and will be formally accepted for publication once it meets all outstanding technical requirements.

Kind regards,

Chiara Poletto

Academic Editor

PLOS ONE

Additional Editor Comments (optional):

Reviewers' comments:

Reviewer's Responses to Questions

**Comments to the Author**

1. If the authors have adequately addressed your comments raised in a previous round of review and you feel that this manuscript is now acceptable for publication, you may indicate that here to bypass the “Comments to the Author” section, enter your conflict of interest statement in the “Confidential to Editor” section, and submit your "Accept" recommendation.

Reviewer #1: All comments have been addressed

Reviewer #2: All comments have been addressed

2. Is the manuscript technically sound, and do the data support the conclusions?

Reviewer #1: Yes

Reviewer #2: Yes

3. Has the statistical analysis been performed appropriately and rigorously? 

Reviewer #1: Yes

Reviewer #2: Yes

4. Have the authors made all data underlying the findings in their manuscript fully available?

Reviewer #1: Yes

Reviewer #2: No

5. Is the manuscript presented in an intelligible fashion and written in standard English?

Reviewer #1: Yes

Reviewer #2: Yes

6. Review Comments to the Author

Reviewer #1: Dear Authors,

all my concerns have been addressed.

I consider the manuscript suitable for publication.

Reviewer #2: The authors did a good job in answering my requests. I am now satisfied with the content of the paper.

However I found that the authors say in their Data Availability Statement that the data are "available upon reasonable request". If the reason behind this choice was that the data cannot be anonymized or shared publicly that should have been explained in the Data Availability Statement. Otherwise the authors are encouraged to share the data on a public repository. Please provide an explanation and change the Statement accordingly.

7. PLOS authors have the option to publish the peer review history of their article (what does this mean?). If published, this will include your full peer review and any attached files.

Reviewer #1: No

Reviewer #2: No

---

## [Editor Report · Acceptance letter]

3 Aug 2022

PONE-D-21-37364R1 

Impact of human mobility and networking on spread of COVID-19 at the time of the 1st and 2nd epidemic waves in Japan: an effective distance approach 

Dear Dr. Nohara:

I'm pleased to inform you that your manuscript has been deemed suitable for publication in PLOS ONE. Congratulations! Your manuscript is now with our production department. 

Kind regards, 

on behalf of

Dr. Chiara Poletto 

Academic Editor

PLOS ONE